# The Relationship between Selected Demographic Factors and Speech Organ Dysfunction in Sporadic ALS Patients

**DOI:** 10.3390/medicina56080390

**Published:** 2020-08-05

**Authors:** Wioletta Pawlukowska, Bartłomiej Baumert, Monika Gołąb-Janowska, Agnieszka Meller, Karolina Machowska-Sempruch, Agnieszka Wełnicka, Edyta Paczkowska, Iwona Rotter, Bogusław Machaliński, Przemysław Nowacki

**Affiliations:** 1Department of Medical Rehabilitation and Clinical Physiotherapy, Pomeranian Medical University, 71-210 Szczecin, Poland; iwrot@wp.pl; 2Department of Neurology, Pomeranian Medical University, 71-252 Szczecin, Poland; monikagj@op.pl (M.G.-J.); agoschorska@gmail.com (A.M.); karolinamachowska88@gmail.com (K.M.-S.); awelnicka@gmail.com (A.W.); przemyslaw.nowacki@pum.edu.pl (P.N.); 3Department of General Pathology, Pomeranian Medical University, 70-111 Szczecin, Poland; bbaumert@pum.edu.pl (B.B.); edyta.paczkowska@pum.edu.pl (E.P.); machalin@pum.edu.pl (B.M.)

**Keywords:** amyotrophic lateral sclerosis, dysarthria, speech disorders, Frenchay dysarthria assessment

## Abstract

*Background and objectives*: Speech disorders are observed in 30% of newly diagnosed sporadic amyotrophic lateral sclerosis (ALS) patients. Characterized by a dynamic course, dysfunction of articulation has not so far been well understood. The aim of this study was to analyze the influence of demographic factors (sex, age, duration of the disease) and concomitant diseases (degenerative spine disease, depression, hypertension, hypothyroidism, hyperthyroidism, and allergy) on the functioning of speech organs in ALS patients. *Materials and Methods*: The study group consisted of 65 patients with sporadic ALS. Patients were examined for articulatory functions by means of the Frenchay Dysarthria Assessment (FDA). *Results*: 68% of the study sample had spinal disorders. Logistic regression analysis showed that a decline in the functioning of lips, soft palate, length of phonation, and voice loudness was more common among men. Patients diagnosed with degenerative spine disease more often suffered from respiratory disorders, while younger patients (<60 years of age) significantly more often had the impairment of the sentence and spontaneous speech functions. *Conclusions*: The male gender in patients with ALS is associated with an increased risk of deterioration of the phonation length function. Patients under 60 years of age are associated with more often pronouncing sentences disorders and spontaneous speech disorders.

## 1. Background

Amyotrophic lateral sclerosis (ALS) is a progressive neurodegenerative disease, leading to speech and swallowing impairments. At the moment of diagnosis, 30% of patients already have bulbar symptoms. While men are more likely to develop ALS (1.5–2:1) [1], women more often present its bulbar form [2]. Recent publications indicate that the bulbar form of ALS is particularly more often observed in older women [3]. Dysarthria disorders in sporadic ALS make a complete loss of the ability to speak quicker [4].

Additionally, bulbar sites of onset and older age at the time of diagnosis predict more rapid disease progression [5]. The incidence of ALS in European countries is higher than in ethnically mixed populations. The mean age of sporadic ALS onset in Europe is 65 years, and 55 years for its genetically determined form. The risk of developing ALS increases in the age bracket of 50–75 years, and then starts to decrease.

The literature on the possible relationship between ALS and risk factors is rather extensive, but the authors of the available studies mainly focused on global motor disorders [2,6,7,8,9,10,11,12]. Reports on the influence of demographic factors and concomitant diseases on articulation disorders in ALS are rare.

The aim of this study was to analyze the influence of demographic factors (sex, age, education, duration of the disease) and concomitant diseases (degenerative spine disease, depression, hypertension, hypothyroidism, hyperthyroidism, and allergy) on the functioning of speech organs in patients with ALS.

## 2. Material and Methods

### 2.1. Patients

The study enrolled 65 patients—27 females and 38 males-aged between 27 and 65 years old (mean: 53.61 ± 9.15) with sporadic ALS [13] according to the El Escorial Revised Criteria [14]. The study was designed as a prospective, open-label, nonrandomized clinical trial in a single center for subjects with ALS. The ALS patients randomly enrolled in the study had to meet the following inclusion criteria:(a)probable or certain sporadic ALS form diagnosed in compliance with the El Escorial Revised Criteria,(b)ability to express informed consent,(c)mild to moderate disability documented by satisfactory bulbar and spinal motor functions (minimum score 3 on the ALS-FRSr scale for swallowing and two points for food preparation and walking-patients able to independently reach the research facility),(d)forced vital capacity (FVC) result ≥ 50%.

The exclusion criteria involved individuals suffering from any concomitant conditions affecting speech quality as well as those aged over 65 years due to the age-related decline in speech [15]. Information on comorbidities was collected based on the history. Diagnosis of comorbidities was made by specialists in the given field.

For designing the experiment, we estimated the required sample size. It was estimated using a sample size calculator. The estimated fraction size was 95% with a significance level of 0.05. The size of the general population of ALS patients in Northwestern Region of Poland amounts to about 90 people. The permissible statistical error was given at the level of 5%. Based on the above formula, the sample size required was 41.

The trial (international number: NCT02193893) was approved by the Ethics Committee of the Pomeranian Medical University in Szczecin, and conducted in accordance with the Declaration of Helsinki [13]. Written informed consent was obtained from each participant. Our study was a secondary analysis of trial data carried out according to the described recruitment rules.

### 2.2. Speech Test—FDA

One of the most important objective tests for evaluating the articulatory organs is the Frenchay Dysarthria Assessment (FDA). The FDA is a standardized test which relies on a 9-point rating scale applied to a patient. It provides information based on the observation of oral structures, functions and speech. The test evaluates the following functions: reflexes, respiration, tongue, lips, the soft palate, larynx, intelligibility. A 5-point rating scale (a–e) is used for the assessment, where letter ‘a’ represents norm, ‘b’ mild severity, ‘c’ moderate severity, ‘d’ considerable severity, ‘e’ profound severity. FDA is also used to assess the severity of the articulatory organs disorders and to monitor the effects of the treatment [16]. The test was conducted by a clinical speech therapist with 12 years of experience of treating neurological conditions, predominantly Parkinson’s disease (PD). The second edition of FDA utilizes the latest findings concerning motor speech disorders and their contribution to a neurological diagnosis. It has good feasibility (missing data < 5%), a high reliability of the total score (0.94), an excellent inter-rater agreement for the total score (0.96) and moderate to large construct validity for 81% of its items [16,17].

Taking into account the severity of articulation disorders according to the FDA scale, each of the domains was divided into two groups: I-without or with mild articulation disorders (‘a’ and ‘b’ disorders according to the FDA); II-with moderate, significant or severe articulation disorders (‘c’, ‘d’ and ‘e’ disorders according to the FDA).

### 2.3. Statistical Analysis

The statistical null hypothesis was: there is no impact of selected demographic factors on speech organs in ALS patients. The alternative hypothesis was: the analyzed demographic factors influenced dysfunction of the articulatory organs in these patients.

To assess the equality of variances for variables, the Levene’s test was used before a comparison of means. The test has shown significance (*p* < 0.05). For this reason and because of the non-normality of the distributions between variables (Shapiro–Wilk test), the numerical data were compared between the groups using the nonparametric Mann–Whitney U-test for variables included to the two groups. The occurrence of nominative clinical data was compared between groups I and II by means of a chi-squared test. To assess the possible influence of examined factors on articulatory organs dysfunction, analysis of logistic regression was used. *p* < 0.05 was considered to indicate statistical significance. All statistical analyses were performed with STATISTICA 12.5 PL.

## 3. Results

General characteristics of the examined population was presented in Table 1.

Table 2 illustrates the severity of speech dysfunction in ALS patients divided into two groups measured by objective means of FDA scale.

As Table 2 shows, moderate, significant, and severe articulation disorders (group II) were significantly more often observed in the domains of phonation time, cough reflex, tongue mobility and palate performance, while words, sentence, and breathing disorders were significantly more often mild or did not occur at all.

Regression analysis was performed to determine the relationship between demographic factors and the severity of speech organ dysfunction. The results are shown in Table 3.

Logistic regression analysis revealed a strong relationship between male sex and the risk of deterioration of length of phonation (OR 4.54; *p* = 0.013), and between a young age and the impairment of the sentence and spontaneous speech functions (OR 1.11; *p* = 0.016 and OR 1.07; *p* = 0.046, respectively). The risk of a decline in the functioning of lips and soft palate, as well as voice loudness, was also higher in men, while the risk of respiratory disorders was higher in patients with degenerative spine disease, however, statistically insignificant.

## 4. Discussion

Dysarthria disorders in ALS are characterized by a different age of onset, as well as varied course and severity. Due to a dynamic pathological process and a wide range of possible symptoms, it seems reasonable to analyze factors that may have an impact on the functioning of speech organs in these patients. Therefore, we decided to analyze the potential influence of selected demographic factors and concomitant diseases on the course of articulation disorders in patients with sporadic ALS. Taking into account the different dynamics of the progression of the articulation organs deficiency in patients with ALS, it is necessary to indicate as many factors as possible predisposing to this dysfunction.

It is worth mentioning that, despite the fact that patients were randomly selected to the study, the population distribution of the recruited patients was very similar to the mean distribution described in the literature [11,18]. The men to women ratio in the study sample was 1.4, and the ratio described in the literature is 1–2 [19,20,21]. The mean age of ALS onset was 53.61 years (while the average is 51–66 years) [18,21,22,23,24], and 68% of the patients had spinal disorders (literature average 58–82%) [25,26,27]. Such a distribution in a relatively large group of patients allows for drawing conclusions and transposing them to the general population of ALS patients.

Our study demonstrated that a decline in the functioning of speech organs (lips and soft palate) as well as in the length of phonation and voice loudness is associated with male sex of ALS patients. The differences were significant in the case of the length of phonation. The literature data seem to confirm differences in the structure of the corticomedullary junction, and especially in neuron excitability, between sexes [28,29,30]. The above-mentioned changes in speech organs usually occur as a precursory symptom of the bulbar onset of ALS. It can be presumed that bulbar changes in men are more dynamic and more severe. Sex-related differences in the functioning of speech organs in ALS patients may result from the fact that the functioning of the twelfth cranial nerve is determined by the expression of the *N*-methyl-*D*-aspartate (NMDA) receptor subunits [31]. Estrogens increase the expression of the NMDA receptor, which stimulates the conduction, hence its possible lower sensitivity in the group of men [32].

Scientific studies conducted so far demonstrated more severe dysarthria and respiratory problems in men with ALS [33]. Sex-dependent differences in the course of ALS can be of different origin. Not without significance is the fact that men are more physically fit, whereas physical activity increases the risk of ALS [12]. Additionally, men more often have occupations that potentially increase the risk of ALS, such as physical work and professions that involve exposure to heavy metals and toxins [34]. Our study confirmed previous reports on the influence of sex on the course of disease.

Better functioning of the organs of articulation in women may also be related to differences in the frontal lobes and organization of brain functions. Women better communicate and process language stimuli [35]. While doing phonological tasks, men activate mainly the left hemisphere, whereas women show bilateral activation [36].

Our study confirmed that degenerative spine disease (preceding ALS onset) is associated with a decline in respiratory function in the course of ALS (OR 3.11; *p* = 0.083). Similar conclusions were drawn by Grolez et al. [37]. The upper and lower motor neurons have a significant impact on respiratory function. Cervical spinal cord (C3–C5) neurons modulate diaphragm motor neurons so that they stimulate the process of respiration and control the blood flow [38]. Degenerative spine disease most probably generates damage to both the upper and lower motor neurons and their processes that are closest to degenerative changes in the cervical spine. Thus, damage on the level of both neurons, caused by degenerative changes in the cervical spine, can speed up the development of respiratory problems and make them worse. This view is shared by other authors [39,40,41,42]. It is also supposed that spinal cord injury causes changes in the motor cortex, controlling diaphragm movements [43].

Logistic analysis also revealed a significant association between articulation disorders (sentence and spontaneous speech) and the age below 60 years. Attention is drawn to the fact that younger individuals are also at a higher risk of developing frontotemporal dementia [44,45,46,47]. It is possible that cognitive function disorders can contribute to the dysfunction of articulation in the domains of sentence and spontaneous speech, which are associated with executive functions and cognitive flexibility [48]. Impairment of this process is associated with changes in the prefrontal cortex [49]. Researchers claim that speech slows down when the bulb becomes affected. The net of junctions between corticomedullary fibers and prefrontal projections plays a key role in executive functions, including spontaneous speech [50].

Based on our study, it can be concluded that bulbar symptoms develop more quickly in patients below 60 years of age. Studies have shown that the quality of speech in ALS deteriorates with the decline in executive functions [51]. As results from our observation, ALS patients with a higher risk of developing speech disorders (i.e., younger men) should be carefully controlled as early as possible and, if necessary, provided with speech-therapy.

The main drawback of the study was the limited range of demographic determinants imposed by strict inclusion criteria and high heterogeneity of the study group.

## 5. Conclusions

The male gender in patients with ALS is associated with an increased risk of deterioration of the phonation length function.Patients under 60 years of age are associated with more often pronouncing sentences’ disorders and spontaneous speech disorders.

## Figures and Tables

**Table 1 medicina-56-00390-t001:** General characteristics of the examined population.

Parameters		All Patients (*n* = 65)
Age (years; mean ± SD)		53.61 ± 9.15
Age (years)	≤60	49 (75.38%)
	>60	16 (24.61%)
Education	higher	20 (30.76%)
	secondary	33 (50.76%)
	vocational	12 (18.46%)
Disease duration (months; mean ± SD)		24.61 ± 22.08
Sex	females	27 (41.53%)
	males	38 (58.46%)
Onset of symptoms	spinal	44 (67.69%)
	bulbar	21 (32.30%)
Comorbidities:		*n* (% rate)
Hypertension		15 (23.07%)
Hypothyroidism/hyperthyroidism		8 (12.30%)
Degenerative spine disease		16 (24.61%)
Depression		10 (15.38%)
Allergy		10 (15.38%)

SD: Standard Deviation.

**Table 2 medicina-56-00390-t002:** Intensity of articulation functions in ALS patients divided into groups based on the FDA scale.

Articulatory Functions	Group I	Group II
Breathing	48 (73.84%)	17 (26.15%)
Cough reflex	36 (55.38%)	29 (44.61%)
Tongue mobility	37 (56.92%)	28 (43.07%)
Swallowing reflex	42 (64.61%)	23 (35.38%)
Lips performance	40 (61.53%)	25 (38.46%)
Palate performance	38 (58.46%)	27 (41.53%)
Voice loudness	42 (64.61%)	23 (35.38%)
Pitch	42 (64.61%)	23 (35.38%)
Phonation time	23 (35.38%)	42 (64.61%)
Control of saliva	42 (64.61%)	23 (35.38%)
Words	55 (84.61%)	10 (15.38%)
Sentence	49 (75.38%)	16 (24.61%)
Spontaneous speech	40 (61.53%)	25 (38.64%)

Groups: I—without or with mild articulation disorders (‘a’ and ‘b’ disorders according to the FDA); II—with moderate, significant or severe articulation disorders (‘c’, ‘d’ and ‘e’ disorders according to the FDA). ALS: amyotrophic lateral sclerosis; FDA: Frenchay Dysarthria Assessment.

**Table 3 medicina-56-00390-t003:** Analysis of logistic regression.

Variables	OR	*p*-Value
	Sex	
Lips	2.6860	0.086
Soft palate	3.1972	0.068
Length of phonation	4.5441	0.013 *
Voice loudness	2.6110	0.073
	age	
Sentence	1.1084	0.016 *
Spontaneous speech	1.0747	0.046 *
	degenerative spine disease	
Breathing	3.1121	0.083 *

* *p* < 0.05. OR: odds ratio.

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
