# Peer review of "The Relationship between Selected Demographic Factors and Speech Organ Dysfunction in Sporadic ALS Patients"

_medicina, 2020, doi:10.3390/medicina56080390_

Round 1
Reviewer 1 Report
- Interesting paper - focus on sociodemographic determinants is interesting and valuable (in theory)
- Patient group is interesting - one of the lesser studied groups - which is a positive - the patient group is also well defined using appropriate criteria
- The sample size seems fairly reasonable - but the authors would be advised to add more information on how the target sample size was calculated and also on participant flow - i.e. what the recruitment rate was like
- I think the range of sociodemographic determinants is too limited - this needs to be clearly noted as a major limitation - the methods section also needs to be much clearer as to what sociodemographic variables were selected and why. There seems to be no section on independent variables, only dependent variables. Socioeconomic status/social class seems a major omission as is level of education.
- The study is described as a non-randomised controlled trial. However, the analysis presented seems to be a cross-sectional analysis and not a trial. This is very unclear - is it secondary analysis of trial data?
- The rationale for this analysis is quite unclear - not helped by the very limited sociodemographic measures and the seemingly cross-sectional nature of the study.
- Conclusions suggest causation in a way that isn't supported by the data nor the study design
Author Response
Medicina Szczecin, July 22, 2020
Editors
Manuscript number: 858460
Dear Editor,
Please, find enclosed our revised paper entitled “The relationship between selected sociodemographic factors and speech organ dysfunction in sporadic ALS patients” by Wioletta Pawlukowska, BartÅ‚omiej Baumert, Monika GoÅ‚Ä…b-Janowska, Agnieszka Meller, Karolina Machowska-Sempruch, Agnieszka WeÅ‚nicka, Edyta Paczkowska, Iwona Rotter, BogusÅ‚aw MachaliÅ„ski, PrzemysÅ‚aw Nowacki. This paper has been prepared in our departments and submitted exclusively to Medicina.
We were pleased to read the constructive comments of the reviewers and their suggestion that the manuscript could be considered for publication in the Journal, with some major revision. We reworked and corrected our paper according to the reviewers’ requests, performing changes in main manuscript. In revised manuscript all changes are indicated using the editing tools. In response to reviewers, all the changes are indicated in red. We trust that the revised version of our manuscript is clearer and strengthened scientifically. We thank the reviewers for their comments and careful evaluation of our paper. We are happy to address all the reviewer comments point by point below.
We hope it will now meet with your approval for publication in Medicina. Thank you for your time and I am looking forward to hearing from you.
Sincerely yours,
Corresponding author:
Wioletta Pawlukowska, PhD
Department of Medical Rehabilitation and Clinical Physiotherapy,
Department of Neurology,
Pomeranian Medical University in Szczecin
Żołnierska 54,
71-210 Szczecin, Poland
phone: +48 914800914
fax: +48 914800918
e-mail: [email protected]
---------------------------------------------- Reviewer #1’s Comments -------------------------------------
Dear Reviewer,
Thank you for your comments and kind opinion concerning our manuscript entitled “The relationship between selected sociodemographic factors and speech organ dysfunction in sporadic ALS patients”. We have studied the comments carefully and have made corrections, which we hope, will meet with your approval.
Points of criticism:
1,2) Interesting paper - focus on sociodemographic determinants is interesting and valuable (in theory). Patient group is interesting - one of the lesser studied groups - which is a positive - the patient group is also well defined using appropriate criteria.
(The response)
Thank you for the positive feedback concerning the selected topic and study group.
- The sample size seems fairly reasonable - but the authors would be advised to add more information on how the target sample size was calculated and also on participant flow - i.e. what the recruitment rate was like.
(The response)
We wish to thank the Reviewer for the helpful comment. According to Reviewer’s comment, we have added information on the calculation of the sample size:
“For designing the experiment, we estimated the required sample size. It was estimated using a sample size calculator. The estimated fraction size was 95% with a significance level of 0.05. The size of the general population of ALS patients in Northwestern Region of Poland amounts circa 90 people. The permissible statistical error was given at the level of 5%. Based on above formula the sample size required was 41.
”
- I think the range of sociodemographic determinants is too limited - this needs to be clearly noted as a major limitation - the methods section also needs to be much clearer as to what sociodemographic variables were selected and why. There seems to be no section on independent variables, only dependent variables. Socioeconomic status/social class seems a major omission as is level of education.
(The response)
We wish to thank the Reviewer for the helpful comment. We selected the most common sociodemographic determinants for the study in order to explore as many correlations as possible. However, the limited number of risk factors results from the limitations during recruitment imposed by the trial inclusion criteria. Additionally, not all obtained results were included in the article. We focused mainly on presenting selected, statistically significant results. There was no correlation between the level of education and the efficiency of articulation organs in patients with ALS. The economic and social status was not analyzed. According to Reviewer’s suggestion we have supplemented Table 1 with information on patients’ education:
Table 1. General characteristics of the examined population
Parameters |
All patients (n = 65) |
||
Age (years; mean ± SD) |
53.61 ± 9.15 |
||
Age (years) |
≤ 60 |
49 (75.38%) |
|
> 60 |
16 (24.61%) |
||
Education |
higher |
20 (30.76%) |
|
secondary |
33 (50.76%) |
||
vocational |
12 (18.46%) |
||
Disease duration (months; mean ± SD) |
24.61 ± 22.08 |
||
Sex |
females |
27 (41.53%) |
|
males |
38 (58.46%) |
||
Onset of symptoms |
spinal |
44 (67.69%) |
|
bulbar |
21 (32.30%) |
||
Comorbidities: |
n (% rate) |
||
Hypertension |
15 (23.07%) |
||
Hypothyroidism / hyperthyroidism |
8 (12.30%) |
||
Degenerative spine disease |
16 (24.61%) |
||
Depression |
10 (15.38%) |
||
Allergy |
10 (15.38%) |
||
Moreover, according to the reviewer’s request, in the revised manuscript, we have pinpointed study limitations as follows:
“The main drawback of the study was the limited range of sociodemographic determinants imposed by strict inclusion criteria and high heterogeneity of the study group.”
- The study is described as a non-randomised controlled trial. However, the analysis presented seems to be a cross-sectional analysis and not a trial. This is very unclear - is it secondary analysis of trial data?
(The response)
Indeed, our study was a secondary analysis of trial data carried out according to the described recruitment rules. In agreement with Reviewer’s suggestion, we have reworked Patients section as follows:
“Our study was a secondary analysis of trial data carried out according to the described recruitment rules.”
- The rationale for this analysis is quite unclear - not helped by the very limited sociodemographic measures and the seemingly cross-sectional nature of the study.
(The response)
We wish to thank the Reviewer for the comment. Taking into account the different dynamics of the course of articulation disorders in patients with ALS, the aim of our study was to select those patients whose risk of rapid progression of articulation dysfunctions is much greater due to the presence of specific risk factors. Hence, we have modified the Discussion section as follows:
“Taking into account the different dynamics of the progression of the articulation organs defficiency in patients with ALS, it is necessary to indicate as many factors as possible predisposing to this dysfunction.”
- Conclusions suggest causation in a way that isn't supported by the data nor the study design.
(The response)
Thank you for the constructive comment. We have reworked Conclusions section to refer only to significant results obtained in the study:
“Conclusions
- The male gender in patients with ALS significantly increases the risk of deterioration of the phonation length.
- Patients under 60 years of age present significantly more often pronouncing sentences and spontaneous speech disorders.”
We would like to thank the referee for the helpful comments and hope that our revised manuscript is now more balanced and better represents our work. We hope that the revised manuscript is acceptable for publication in Medicina.
---------------------------------------------- Reviewer #2’s Comments -------------------------------------
Dear Reviewer,
We would like to thank for careful and thorough reading of this manuscript and for the thoughtful comments and constructive suggestions, which help to improve the quality of this manuscript. Our response follows the comments.
Points of criticism:
- In the section about statistical analysis, the authors write: “For this reason and because of the non-normality of the distributions between variables (Shapiro–Wilk’s test)”. This sentence seems to be incomplete.
(The response)
We wish to thank the Reviewer for the constructive comment. We have completed the missing sentence:
“For this reason and because of the non-normality of the distributions between variables (Shapiro–Wilk test), the numerical data were compared between the groups using the nonparametric Mann–Whitney U-test for variables included to the two groups.”
- In the discussion, the authors write: “Our study demonstrated a decline in the functioning of speech organs (lips and soft palate) as well as in the length of phonation and voice loudness in the group of men with ALS”. I would mention only statistically significant results in this section.
(The response)
We changed the results and included only statistically significant data.
- In the discussion, the authors write: “Based on our study, it can be concluded that bulbar symptoms develop more quickly in patients below 60 years of age, leading to the dysfunction of executive functions, which directly affects the quality of speech”. The results do not allow to conclude that bulbar symptoms lead to the deterioration of executive functions. This concept should be more clearly expressed.
(The response)
We wish to thank the Reviewer for the supportive comment. According to the advice, we have modified the Discussion section as follows:
“Based on our study, it can be concluded that bulbar symptoms develop more quickly in patients below 60 years of age. Studies have shown that the quality of speech in ALS deteriorates with the decline in executive functions [51].”
- In the conclusions, the authors write: “Male sex and young age of ALS patients substantially increase the risk of progression of the impairment of speech organs (such as lips, soft palate) and functions (length of phonation, voice loudness, sentence, spontaneous speech)”. This concept should actually refer only to statistically significant associations.
(The response)
We wish to thank the Reviewer for the practical comment. We have modified the Conclusions section to be consistent with the results:
„Conclusions
- The male gender in patients with ALS significantly increases the risk of deterioration of the phonation length function.
- Patients under 60 years of age present significantly more often pronouncing sentences and spontaneous speech disorders.
We would like to thank the referee for the helpful comments and hope that our revised manuscript is now more balanced and better represents our work. We hope that the revised manuscript is acceptable for publication in Medicina.
Reviewer 2 Report
This is an interesting study on sociodemographic variables and comorbidities which can influence deterioration of speech deficits in ALS patients.
Although the results do not represent a major advance in understanding disease mechanisms and are not supposed to change clinical practice substantially, they can improve some aspects of monitoring and, if possible, of rehabilitation of dysarthria in ALS.
I recommend that the authors distinguish more clearly, especially in the discussion and conclusions, statistically significant results from non-significant ones. The first ones can indeed be presented to support specific pathophysiological hypotheses, as the authors do in the discussion; on the contrary, non-significant results do not allow similar conclusions.
The following comments refer to specific parts of the text.
- In the section about statistical analysis, the authors write: “For this reason and because of the non-normality of the distributions between variables (Shapiro–Wilk’s test)”. This sentence seems to be incomplete.
- In the discussion, the authors write: “Our study demonstrated a decline in the functioning of speech organs (lips and soft palate) as well as in the length of phonation and voice loudness in the group of men with ALS”. I would mention only statistically significant results in this section.
- In the discussion, the authors write: “Based on our study, it can be concluded that bulbar symptoms develop more quickly in patients below 60 years of age, leading to the dysfunction of executive functions, which directly affects the quality of speech”. The results do not allow to conclude that bulbar symptoms lead to the deterioration of executive functions. This concept should be more clearly expressed.
- In the conclusions, the authors write: “Male sex and young age of ALS patients substantially increase the risk of progression of the impairment of speech organs (such as lips, soft palate) and functions (length of phonation, voice loudness, sentence, spontaneous speech)”. This concept should actually refer only to statistically significant associations.
Author Response
Dear Reviewer,
We would like to thank for careful and thorough reading of this manuscript and for the thoughtful comments and constructive suggestions, which help to improve the quality of this manuscript. Our response follows the comments.
Points of criticism:
- In the section about statistical analysis, the authors write: “For this reason and because of the non-normality of the distributions between variables (Shapiro–Wilk’s test)”. This sentence seems to be incomplete.
(The response)
We wish to thank the Reviewer for the constructive comment. We have completed the missing sentence:
“For this reason and because of the non-normality of the distributions between variables (Shapiro–Wilk test), the numerical data were compared between the groups using the nonparametric Mann–Whitney U-test for variables included to the two groups.”
- In the discussion, the authors write: “Our study demonstrated a decline in the functioning of speech organs (lips and soft palate) as well as in the length of phonation and voice loudness in the group of men with ALS”. I would mention only statistically significant results in this section.
(The response)
We changed the results and included only statistically significant data.
- In the discussion, the authors write: “Based on our study, it can be concluded that bulbar symptoms develop more quickly in patients below 60 years of age, leading to the dysfunction of executive functions, which directly affects the quality of speech”. The results do not allow to conclude that bulbar symptoms lead to the deterioration of executive functions. This concept should be more clearly expressed.
(The response)
We wish to thank the Reviewer for the supportive comment. According to the advice, we have modified the Discussion section as follows:
“Based on our study, it can be concluded that bulbar symptoms develop more quickly in patients below 60 years of age. Studies have shown that the quality of speech in ALS deteriorates with the decline in executive functions [51].”
- In the conclusions, the authors write: “Male sex and young age of ALS patients substantially increase the risk of progression of the impairment of speech organs (such as lips, soft palate) and functions (length of phonation, voice loudness, sentence, spontaneous speech)”. This concept should actually refer only to statistically significant associations.
(The response)
We wish to thank the Reviewer for the practical comment. We have modified the Conclusions section to be consistent with the results:
„Conclusions
- The male gender in patients with ALS significantly increases the risk of deterioration of the phonation length function.
- Patients under 60 years of age present significantly more often pronouncing sentences and spontaneous speech disorders.
We would like to thank the referee for the helpful comments and hope that our revised manuscript is now more balanced and better represents our work. We hope that the revised manuscript is acceptable for publication in Medicina.
Round 2
Reviewer 1 Report
I consider the manuscript much improved, but there remain 3 major issues to resolve:
- Is 'sociodemographic' an appropriate term to describe the determinants being explored, given that no socioeconomic variables are available?
- The authors still need to be more careful about claims of causation using these limited cross-sectional data - for example 'increases' should be 'is associated with an increase in' - please check carefully
- The authors' decision to only show statistically significant findings could be misleading in terms of the overall picture of the results
Author Response
Medicina Szczecin, July 28, 2020
Editors
Manuscript number: 858460
Dear Editor,
Please, find enclosed our revised paper entitled “The relationship between selected demographic factors and speech organ dysfunction in sporadic ALS patients” by Wioletta Pawlukowska, BartÅ‚omiej Baumert, Monika GoÅ‚Ä…b-Janowska, Agnieszka Meller, Karolina Machowska-Sempruch, Agnieszka WeÅ‚nicka, Edyta Paczkowska, Iwona Rotter, BogusÅ‚aw MachaliÅ„ski, PrzemysÅ‚aw Nowacki. This paper has been prepared in our departments and submitted exclusively to Medicina.
We were pleased to read the constructive comments of the reviewers and their suggestion that the manuscript could be considered for publication in the Journal, with some major revision. We reworked and corrected our paper according to the reviewers’ requests, performing changes in main manuscript. In revised manuscript all changes are indicated using the editing tools. In response to reviewers, all the changes are indicated in red. We trust that the revised version of our manuscript is clearer and strengthened scientifically. We thank the reviewers for their comments and careful evaluation of our paper. We are happy to address all the reviewer comments point by point below.
We hope it will now meet with your approval for publication in Medicina. Thank you for your time and I am looking forward to hearing from you.
Sincerely yours,
Corresponding author:
Wioletta Pawlukowska, PhD
Department of Medical Rehabilitation and Clinical Physiotherapy,
Department of Neurology,
Pomeranian Medical University in Szczecin
Żołnierska 54,
71-210 Szczecin, Poland
phone: +48 914800914
fax: +48 914800918
e-mail: [email protected]
---------------------------------------------- Reviewer #1’s Comments -------------------------------------
Dear Reviewer,
Thank you for your comments and kind opinion concerning our manuscript entitled “The relationship between selected demographic factors and speech organ dysfunction in sporadic ALS patients”. We have studied the comments carefully and have made corrections, which we hope, will meet with your approval.
Points of criticism:
- Is 'sociodemographic' an appropriate term to describe the determinants being explored, given that no socioeconomic variables are available?
(The response)
We wish to thank the Reviewer for the constructive comment. We agree with the Reviewer, therefore we have modified the title and several sentences in the manuscript as follows:
“The relationship between selected sociodemographic factors and speech organ dysfunction in sporadic ALS patients”
“The aim of this study was to analyze the influence of sociodemographic factors (sex, age, education, duration of the disease) and concomitant diseases (degenerative spine disease, depression, hypertension, hypothyroidism, hyperthyroidism, and allergy) on the functioning of speech organs in patients with ALS.”
“Reports on the influence of sociodemographic factors and concomitant diseases on articulation disorders in ALS are rare.”
“The statistical null hypothesis was: there is no impact of selected sociodemographic factors on speech organs in ALS patients.”
“The alternative hypothesis was: the analysed sociodemographic factors influenced dysfunction of the articulatory organs in these patients.”
“Regression analysis was performed to determine the relationship between sociodemographic factors and the severity of speech organ dysfunction. The results are shown in Table 3.”
“Therefore, we decided to analyze the potential influence of selected sociodemographic factors and concomitant diseases on the course of articulation disorders in patients with sporadic ALS.”
“The main drawback of the study was the limited range of sociodemographic determinants imposed by strict inclusion criteria and high heterogeneity of the study group.”
- The authors still need to be more careful about claims of causation using these limited cross-sectional data - for example 'increases' should be 'is associated with an increase in' - please check carefully.
(The response)
Thank You for another important remark. According to the Reviewers’ suggestion we have modified the Abstract, Discussion and Conclusions section as follows:
„The male gender in patients with ALS significantly increases the risk is associated with an increased risk of deterioration of the phonation length function. Patients under 60 years of age present significantly are associated with more often pronouncing sentences disorders and spontaneous speech disorders.”
„Our study demonstrated that a decline in the functioning of speech organs (lips and soft palate) as well as in the length of phonation and voice loudness is associated with male sex of ALS patients in the group of men with ALS.”
“Our study confirmed the effect of that degenerative spine disease (preceding ALS onset) on a is associated with a decline in respiratory function in the course of ALS (OR 3.11; p = 0.083).”
„1. The male gender in patients with ALS is associated with an increased significantly increases the risk of deterioration of the phonation length function.
- Patients under 60 years of age are associated with present significantly more often pronouncing sentences disorders and spontaneous speech disorders.”
- The authors' decision to only show statistically significant findings could be misleading in terms of the overall picture of the results.
(The response)
We wish to thank the Reviewer for the constructive comment. Indeed, in our study, we have only quoted statistically significant or borderline results. Lack of statistical significance in the population under study means that the obtained results may be random, and thus clinically irrelevant. Additionally, we decided that presentation of very complex tables with logistic regression analysis would reduce the transparency of the work.
We hope that we have provided clear and concise answers to all of the Reviewer’s questions and that our manuscript is now more detailed and constitutes a better representation of our work what makes it suitable for publication in Medicina.
Round 3
Reviewer 1 Report
I am now satisfied with the revisions